# Protective Efficacy of an mRNA Vaccine Against HP-PRRSV Challenge in Piglets

**DOI:** 10.3390/microorganisms13061332

**Published:** 2025-06-07

**Authors:** Jiaqi Liu, Shiting Ni, Yaning Lv, Ze Tong, Pingxuan Liu, Xin Zong, Guosheng Chen, Yan Zeng, Chenchen Wang, Chen Tan

**Affiliations:** 1National Key Laboratory of Agricultural Microbiology, College of Veterinary Medicine, Huazhong Agricultural University, Wuhan 430070, China; ljq2021302010149@webmail.hzau.edu.cn (J.L.); nissa@webmail.hzau.edu.cn (S.N.); lvyaning@webmail.hzau.edu.cn (Y.L.); tongze3375@webmail.hzau.edu.cn (Z.T.); liupx@webmail.hzau.edu.cn (P.L.); zongx@webmail.hzau.edu.cn (X.Z.); chenguosheng@webmail.hzau.edu.cn (G.C.); zengyanyan@webmail.hzau.edu.cn (Y.Z.); 2018302110164@webmail.hzau.edu.cn (C.W.); 2Key Laboratory of Preventive Veterinary Medicine in Hubei Province, The Cooperative Innovation Center for Sustainable Pig Production, Wuhan 430070, China

**Keywords:** PRRSV, pathogenicity analysis, antigenic epitopes, mRNA vaccine, immune protection

## Abstract

The global pork production sector continues to experience substantial financial burdens attributable to porcine reproductive and respiratory syndrome virus (PRRSV) infections. Despite the current epidemiological landscape in which NADC30-like strains predominate alongside cocirculating diverse PRRSV subtypes, highly pathogenic PRRSV (HP-PRRSV) remains a persistent threat. Furthermore, currently available commercial PRRS vaccine formulations exhibit restricted heterologous protection efficacy. The development of novel mRNA-based vaccines represents a promising strategy for PRRS mitigation protocols. In response to these epidemiological challenges, an HP-PRRSV strain (Lineage 8), designated as JX021, was isolated and characterized in this study. Pathogenicity experiments confirmed that JX021 induces severe clinical symptoms in piglets. Moreover, by combining immunoinformatics and literature-guided approaches, critical antigenic epitopes on HP-PRRSV (represented by the JXA1 strain) structural proteins were identified, enabling the design and synthesis of a multiepitope mRNA vaccine. The survival of piglets immunized with the mRNA vaccine was higher than that of the inactivated vaccine immunization group and the PBS group. Compared with the inactivated vaccine group, the mRNA vaccine group presented reductions in viremia and lung lesions. These findings provide new insights into the design and development of further PRRS vaccine research.

## 1. Introduction

Porcine reproductive and respiratory syndrome virus (PRRSV) is recognized as the etiological agent responsible for a clinically significant swine pathogen of substantial concern. Clinically, PRRS has a distinct biphasic clinical presentation marked by gestational complications in breeding females and pulmonary distress across neonatal and juvenile swine cohorts, resulting in devastating economic losses [1]. Two genotypes are recognized within PRRSV: genotype 1 (PRRSV-1), which is epitomized by the Lelystad virus, and genotype (PRRSV2), which is typified by ATCC VR-2332 and serves as a reference strain for molecular characterization [2,3]. PRRSV-2 comprises nine phylogenetic lineages, four of which exhibit epidemiological dominance in Chinese swine populations: Lineage 1 (NADC30-like), Lineage 3 (QYYZ-like), Lineage 5 (VR-2332-like), and Lineage 8 (CH-1a-like and JXA1-like) [4,5]. In 2006, the emergence of HP-PRRSV strains exhibiting noncontiguous deletions (1 aa and 29 aa) in the NSP2 coding region which are associated with severe swine infections was first documented in the Chinese swine population [6,7]. Following their initial emergence post-2013, NADC30-like PRRSV strains became predominant in China [8]. In 2018, the initial detection of NADC34-like variants was documented within Chinese swine herds [9], whereas PRRSV-1 (European genotype) strains have increasingly spread to multiple regions in recent years [10], posing challenges to PRRS prevention.

PRRSV is an enveloped, single-stranded positive-sense RNA virus classified within the *Arteriviridae* family [11]. The viral genome comprises approximately 15 kilobases, harboring 11 defined open reading frames. ORF1a/1b encodes polyproteins essential for the viral replication machinery, whereas ORFs 2–7 coordinate the expression of structural components (GP2-GP5, M, E, and N proteins) which mediate viral entry, assembly, and immune evasion [12,13,14]. B-cell epitopes elicit neutralizing antibody generation through immunogenic activation, which is critical for controlling viral dissemination. T-cell epitopes, particularly cytotoxic T lymphocyte (CTL) epitopes, elicit cellular immune responses and confer cross-protection [15]. The major structural proteins of PRRSV contain abundant antigenic epitopes, such as glycosylated and immunodominant hypervariable decoy epitopes on the PRRSV-2 GP5 protein [16]. Therefore, these proteins serve as primary immunogens and ideal antigenic targets for PRRSV vaccine development.

The unprecedented outbreak of COVID-19 has dramatically accelerated mRNA vaccine development [17], indicating significant promise for advancing comprehensive prevention and control strategies against PRRS. mRNA vaccines offer distinct advantages, including a low risk profile, potent immunogenicity, and rapid development timelines [18]. To date, mRNA-based platforms have demonstrated remarkable potential in combating viral diseases other than COVID-19 [19,20]. Notably, this development has opened new avenues for preventing swine infectious diseases [21]. Based on this rationale, we employed a lipid nanoparticle-encapsulated mRNA (mRNA-LNP) platform to develop a PRRS mRNA vaccine candidate.

In this study, we isolated and sequenced a novel HP-PRRSV strain (JX021) from China and characterized the virulence in piglets through challenge experiments. We then constructed an mRNA vaccine candidate against PRRSV encoding a multi-epitope polypeptide by screening the important B- and T-cell epitopes of PRRSV structural proteins. The immunogenicity of the mRNA vaccine and its capacity to protect against HP-PRRSV challenge were evaluated in comparison with a commercial inactivated vaccine against PRRSV in vivo. The results provide more insights for the prevention and control of PRRS.

## 2. Materials and Methods

### 2.1. Animals and Ethics Statement

The piglets (*n* = 28, 28 ± 3 days old) were sourced from a PRRSV-free pig farm in Wuhan, Hubei. RT-qPCR and ELISA were performed to detect PRRSV antigens and antibodies in piglets, which were negative. All experimental procedures involving the animals in this study were performed under protocols approved by the Scientific Ethics Committee of Huazhong Agricultural University. The approval code is HZAUSW-2024-0054.

### 2.2. Cells and Reagents

293T and Marc-145 cells were obtained from the China Center for Type Culture Collection (Wuhan, China). Dulbecco’s modified Eagle’s medium (DMEM) (Gibco, Langley, OK, USA) supplemented with 10% fetal bovine serum (FBS) (Gibco, Langley, OK, USA) in a 5% CO_2_ incubator was used for cell culture. PAMs were isolated from the lungs of healthy piglets at 4 weeks of age and cultured with RPMI-1640 (Gibco, Langley, OK, USA) supplemented with 10% FBS.

### 2.3. Sample Source and Virus Isolation

We isolated a PRRSV strain from lung samples from a farm in Jiangxi Province on 21 August 2022. The lung tissues were ground, and RNA was extracted, followed by detection of PRRSV via RT-qPCR. The remaining ground solution was inoculated onto Marc-145 cells as well as PAMs for PRRSV isolation. After 48 h, the cultures were harvested and stored as virus stocks at −80 °C. After plaque purification, the 5th passage culture of Marc-145 cells was used for genome sequencing and animal experiments. Prior to viral inoculation, Marc-145 cell monolayers were established in 96-well culture plates through 12-h preculture incubation. The supernatant from the plaque-purified cell culture was subjected to a logarithmic dilution series (10^−1^ to 10^−8^) using DMEM as the diluent, with 100 μL added to each well. Eight wells were replicated per dilution. Then, the cells were incubated for 48 h, and the 50% tissue culture infective dose (TCID_50_) was calculated using the Reed–Muench method in an immunofluorescence assay (IFA). We named the isolated strain JX021.

### 2.4. Genome Cloning and Sequencing

PRRSV-positive nucleic acids were reverse-transcribed to obtain cDNA. The PRRSV ORF5 and NSP2 genes and the whole genome were amplified and sequenced. The sequences of primers used were as follows: PRRSV ORF5-F: ACCTGAGACCATGAGGTGGG; PRRSV ORF5-R: GCAAGCACAAACGGCATCTG; PRRSV NSP2-F: ATGTTGTGCTTCCTGGGGTTG; and PRRSV NSP2-R: CTTGACAGGGAGCTGCTTGA. The whole-genome primers were developed according to prior research [22]. The whole-genome sequence of JX021 was deposited in GenBank under the accession number PV467459. Phylogenetic analysis was performed using MEGA11 (Molecular Evolutionary Genetics Analysis version 11) software, and prior research was based on the neighbor-joining method [23]. The reference sequences of PRRSV were as followed (Table 1).

### 2.5. mRNA-Based Vaccine

The structural proteins GP2a, GP3, GP4, GP5, M, and N of the JXA1 strain (GenBank accession number EF112445.1), which is the representative strain of the HP-PRRSV (Lineage 8), were the target proteins for the mRNA ORF regions. First, the linear B-cell epitopes of the structural proteins were predicted. To improve the prediction accuracy, three web-based tools—ABCpred (https://webs.iiitd.edu.in/raghava/abcpred/ABC_submission.html; Access: 30 April 2024), BepiPred-3.0 (https://services.healthtech.dtu.dk/services/BepiPred-3.0/; Access: 30 April 2024), and IEDB.org (http://tools.iedb.org/main/bcell/; Access: 30 April 2024)—were comprehensively utilized. The iBCE-EL server (http://thegleelab.org/iBCE-EL/iBCE.html; Access: 30 April 2024) was subsequently employed to validate the predicted epitopes.

T-cell epitope prediction tools are based on the processes of peptide fragment binding to the major histocompatibility complex (MHC), intracellular processing, and transport. These tools calculate predictive scores for each peptide as a potential T-cell epitope using this information. Leukocyte antigens (LAs), also known as MHC, are classified into class I and class II antigens, which activate cytotoxic T lymphocytes (CTLs) and helper T lymphocytes (HTLs), respectively. To increase prediction accuracy, the servers NetMHCpanEL4.1 (http://tools.iedb.org/mhci/; Access: 30 April 2024) and NetMHCpan4.1 (https://services.healthtech.dtu.dk/; Access: 30 April 2024) were employed for comprehensive prediction of CTL candidate epitopes targeting GP2a, GP3, GP4, GP5, and M. The threshold for swine alleles was set to near 0, with epitope lengths ranging from 8 to 14 residues. Peptides scoring in the top 1% were selected as potential candidates for peptides of the vaccine. The MHC-I Immunogenicity server (http://tools.iedb.org/immunogenicity/; Access: 30 April 2024) in the IEDB database was subsequently used to filter epitopes with immunogenicity scores > 0.2 as final predictions. For HTL candidate epitope prediction, the NetMHCIIpanv4.0 EL server (http://tools.iedb.org/mhcii/; Access: 30 April 2024) was utilized to predict peptide binding to MHC II molecules for the structural proteins GP2a, GP3, GP4, GP5, M, and N. The epitope length was set to 15 residues, with candidate epitopes filtered based on the top 10th percentile rank. Based on the predicted structures from NetMHCIIpanv4.0 EL, further analysis was conducted to evaluate the ability of the candidate epitopes to induce IFN-γ and their cytokine activity profiles.

Based on the epitopes obtained from the aforementioned immunoinformatics analyses, the B-cell, HTL, and CTL epitopes were connected sequentially using distinct linkers. To increase the immunogenicity of the candidate vaccine, ANTIGENpro (https://scratch.proteomics.ics.uci.edu/; Access: 30 April 2024) was used to predict the immunogenicity of the constructed candidate vaccine’s ORF region. The ExPASy ProtParam tool (https://web.expasy.org/protparam/; Access: 30 April 2024) was used to predict the physicochemical properties of the candidate vaccine, including the molecular weight, isoelectric point (pI), half-life, and stability. These web-based tools were combined with DNAStar Lasergene 7 Protean (DNASTAR, Madison, WI, USA) to further analyze the immunogenicity, hydrophilicity, and secondary structural features of the screened sequence.

The full PRRS mRNA sequence consists of the following components in the 5′-3′ order: promoter sequence—5′ UTR—Kozak sequence—signal peptide coding sequence—ORF region—3′ UTR-poly(A) tail. The complete sequence was synthesized into mRNA-LNPs (lipid nanoparticles) by Jiangsu Synthgene Biotechnology Co., Ltd. (Nanjing, China).

### 2.6. The Virulence of the JX021 Strain in Pigs

Four-week-old piglets (*n* = 8) were confirmed to be negative for PRRSV antigen and antibodies via RT-qPCR as well as ELISA. The piglets were randomly divided into two groups (*n* = 4): the JX021-challenged group, which received HP-PRRSV JX021 diluted to 10^5^ TCID_50_ via the intranasal route (1 mL) and intramuscular route (1 mL), and the negative control group, which received an equivalent volume of DMEM. Body temperature, weight, and clinical symptoms (scored based on the reported literature [24]) were monitored daily. Post-challenge survival rates were assessed on predetermined Day 14 through daily monitoring. Blood samples were collected at 0, 4, 7, 10, and 14 days post challenge (dpc) to quantify the presence of PRRSV in the blood by RT-qPCR, which is known as viremia. Serum samples were collected at 0, 4, 7, 10, and 14 days post challenge (dpc) to detect the level of PRRSV N antibody by using the commercial ELISA kit (Guangzhou Yoyoung Biology Co., Ltd., Guangzhou, China). A sample-to-positive (S/P) ratio of the piglets equal to or higher than 0.4 is considered positive (cut-off value), which was defined as seroconversion positivity. Tissues were harvested at 14 dpc, and the viral load was detected. The lungs were subjected to histopathological examination.

### 2.7. Vaccination and JX021 Challenge in Pigs

Prior to vaccination, 4-week-old piglets (*n* = 20) were confirmed to be negative for the PRRSV antigen and antibodies via RT-qPCR and ELISA. The piglets were randomly divided into four groups (*n* = 5): the PRRS mRNA-LNP group, inactivated vaccine control group, JX021-challenged control group (PBS group), and negative control group. The commercial inactivated vaccine was based on a PRRSV CH-1a strain and belonged to Lineage 8. Both vaccine groups received primary immunization at 0 d and a booster at 14 days post vaccination (dpv). The control groups were administered PBS at the same time. Blood samples were collected from each group at 14 and 28 dpv to measure PRRSV antibody levels and neutralizing antibody titers in the serum. At 28 dpv, piglets were challenged with HP-PRRSV JX021, which was diluted to 10^5^ TCID_50_, via the intranasal route (1 mL) and intramuscular route (1 mL). The vaccination and challenge experimental groups were as followed (Table 2). Rectal temperature, body weight, and clinical symptoms (scored) were recorded daily post challenge. Survival rates were calculated at 14 days post challenge (dpc). Serum samples were collected at 0, 7, 10, and 14 dpc to quantify the viral load. At 14 days post challenge, tissue samples were collected for virological quantification, and lung tissues were harvested for histopathological examination.

### 2.8. Immunofluorescence Assay (IFA)

One hundred microliters of viral supernatant was inoculated into a pre-seeded cell plate for 1 h. After 36–48 h of incubation, the cells were fixed with 100% methanol at −20 °C for 30 min. The cells were blocked with PBS containing 5% BSA for 1 h and incubated with an anti-PRRSV N protein primary antibody (1:200) at 37 °C for 1 h. A PRRSV N polyclonal antibody was prepared in our laboratory. Cellular cultures were exposed to fluorescein isothiocyanate-conjugated caprine-derived anti-rabbit IgG under standardized incubation protocols (1:500) (Biosharp, Hefei, China) for 30 min at 37 °C, and the cells were observed using a fluorescence microscope (EVOS M7000 Cell Imaging Systems Service Plans, Thermo Fisher Scientific Inc., Erlangen, Germany).

### 2.9. Virus Neutralization Assay

The endpoint neutralization test was used to determine the neutralization titer of the serum. We adopted the method of diluting the serum with fixed virus titer. Marc-145 cells were distributed into 96-well plates at 1 × 10^4^ cells/well and cultured until they reached 90–95% confluence. Neutralization assays were grouped into the following groups: the mRNA vaccine, inactivated vaccine, challenged control, and negative control groups. The obtained serum was pretreated at 56 °C for 30 min for inactivation purposes. The mixture was shaken evenly every 10 min and centrifuged at 5000 rpm (2348× *g*) for 10 min after inactivation. The supernatant was subsequently removed and diluted 2-fold (1:2, 1:4, 1:8, 1:1:16, 1:32, 1:64, 1:128, and 1:256). PRRSV JX021 was diluted to a 100 TCID_50_/50 μL concentration with DMEM, with an initial TCID_50_ of 10^−6.02^/100 μL. The diluted serum was completely mixed with the virus at a ratio of 1:1 for 2 h. The virus–serum mixture was added, and the DMEM supplemented with 2% FBS was replaced every 5–7 days. The cytopathic effect (CPE) was observed for 5–7 days until the results were stable. Under the condition that the control was established, the titers of the specific neutralizing antibody against PRRSV in the tested serum were calculated by using the Reed–Muench’s method, and the highest dilution multiple of the serum that could completely protect the cells from lesion was taken as the neutralizing titer of the tested serum sample.

### 2.10. Histopathological Examination

The lung tissues of the piglets were aseptically collected at 14 dpc and immediately immersed in 10% neutral buffered formalin (NBF, Sigma-Aldrich, Shanghai Trading Co., Ltd., Shanghai, China) for a minimum of 24 h of fixation. For standardized paraffin embedding and microtome sectioning, uniformly trimmed tissue blocks were subsequently sent to Wuhan Servicebio Technology Co., Ltd. (Wuhan, China).

### 2.11. Statistical Analysis

The data in this study are presented as the means ± standard deviations (error bars) for each group. The intergroup differences in the data were analyzed using a one-way analysis of variance (ANOVA) with GraphPad Prism version 8 (GraphPad, Inc., San Diego, CA, USA) and the independent sample *t*-test with IBM SPSS Statistics for Windows, version 27.0 (IBM Corp., Armonk, NY, USA). All the data were analyzed for statistical significance. Ns represents *p* > 0.05. * denotes *p* < 0.05. ** indicates *p* < 0.01. *** signifies *p* < 0.001.

## 3. Results

### 3.1. Isolation and Identification of JX021

PRRSV was detected by RT-qPCR in the lungs of the submitted pigs. Pulmonary tissue homogenates were aseptically seeded into confluent Marc-145 cell monolayers and PAMs for viral propagation for 3–4 days. The results indicate that the JX021 PRRSV strain induced significant cytopathic effects (CPEs) in PAMs as well as Marc-145 cells after 48 h of cultivation (Figure 1A). Moreover, the negative control cells remained healthy with clear cell boundaries. To verify the infection of the PRRSV strain isolated from the lung sample, IFA and Western blotting were employed to detect PRRSV infection in Marc-145 cells cultured for 48 h, with the PRRSV N protein polyclonal antibody used as the primary antibody. As shown in Figure 1B,C, we successfully demonstrated the colonization of Marc-145 cells by JX021. The stability of JX021 PRRSV after the first five passages was assessed by PCR (Figure 1D). After plaque purification of JX021 PRRSV, the TCID_50_ was determined to be 10^−6.02^/100 μL according to the Reed–Muench method. Transmission electron microscopy revealed that the PRRSV virions were spherical with a diameter of approximately 50 nm (Figure 1E). Based on these identification results, JX021 PRRSV was successfully isolated in this study.

### 3.2. Genomic Characteristics of JX021 PRRSV

ORF5-encoded glycoprotein 5 (GP5) constitutes a major virion structural component of PRRSV and serves as the immunodominant antigen driving host-neutralizing immunoglobulin responses [25]. NSP2, conversely, represents the most genetically divergent nonstructural (NS) protein among PRRSV isolates [26]. Therefore, analyzing the genetic evolution of the ORF5 and NSP2 genes of PRRSV is beneficial for understanding the genomic characteristics of the JX021 strain. Phylogenetic evaluation was undertaken by comparing the ORF5 and NSP2 genes and the whole genome with reference sequences. As illustrated in Figure 2A–C, the JX021 strain is a representative of PRRSV-2 and is genetically closest to the JXA1 strain, which is classified as a member of sublineage 8.7 (HP-PRRSV). Additionally, a recombination assessment of the complete genomic sequence of JX021 was executed using SimPlot (version 3) and RDP (version 4), revealing that the JX021 strain exhibits no recombination and is a pure sublineage 8.7 (HP-PRRSV) strain (Figure 2D).

### 3.3. Pathogenicity Analysis of JX021 PRRSV

Piglets (*n* = 4) were challenged with JX021 PRRSV (2 × 10^5^ TCID_50_), and the negative controls (*n* = 4) were inoculated with DMEM. The temperature, body weight, and clinical symptoms of the piglets were monitored for 14 days. After being challenged with JX021, the infected group of piglets presented persistent high fever (40.0–41.9 °C) from 1 to 10 days post challenge (dpc). Their body temperatures returned to below 40 °C after 10 dpc (Figure 3A). The challenged group continued to experience weight loss, whereas the control cohort demonstrated progressive body mass gain (Figure 3B). The challenged piglets displayed various symptoms following challenge, including anorexia, depression, rapid breathing, coughing, shivering, and paralysis. The clinical scores for these symptoms are presented in Figure 3C. Notably, two piglets from the challenged group died at 10 dpc. The overall survival rate of the infected groups was 50% (Figure 3D). Blood sera were obtained at 0, 4, 7, 10, and 14 dpc. The sample-to-positive (S/P) ratios were greater than 0.4 after 7 dpc, as shown in Figure 3E. PRRSV was detectable in nasal swabs at 4 dpc, as shown in Figure 3F. The PRRSV viral load in blood, which is known as PRRSV viremia in the challenged piglets, was detected at 4 dpc and reached a maximum at 10 dpc (Figure 3G). As shown in Figure 3H, PRRSV had the highest copy number in the lungs, followed by effective replication in the lymph nodes and hearts. Importantly, PRRSV was not detected in nasal swabs, blood, or tissues from any of the piglets in the control group. These data collectively suggest that JX021 is highly pathogenic in piglets.

At necropsy, the infected pigs exhibited severe lung consolidation and necrosis. Histomorphological assessment was performed to quantify JX021-induced microanatomical lesions systematically through semiquantitative scoring criteria. As shown in Figure 4, interstitial pneumonia was characterized by thickening of the alveolar septa, as pointed out by the black arrow. Hemorrhagic manifestations were present predominantly within the PRRSV-infected cohorts, along with the infiltration of mononuclear cells, as pointed out by the red arrow. In contrast, no abnormalities were observed in the control group.

### 3.4. Design of PRRS mRNA Vaccine

Several PRRSV structural proteins containing abundant epitopes can induce neutralizing antibodies, among which those targeting the GP5 protein show the most potent antiviral activity in preventing infection [27]. With the aim of developing an mRNA vaccine against HP-PRRSV, the selection of antigens relies mainly on the prediction and screening of antigenic epitopes associated with the structural proteins GP2a, GP3, GP4, GP5, M, and N based on immunoinformatics and previous reports. As presented in Table 3, combined with the predictions of ABCpred, IEDB, and iBCE-EL, the final numbers of linear B-cell epitopes obtained for HP-PRRSV GP2a, GP3, GP4, GP5, M, and N were 2, 3, 3, 5, 4, and 4, respectively.

Using the NetMHCpan4.1 EL and MHC-I immunogenicity tools from the IEDB database and considering their correlation with IFN-γ release, the T-cell epitopes obtained for HP-PRRSV GP2, GP3, GP4, GP5, M, and N were 9, 2, 3, 2, 5, and 4, respectively, as shown in Table 4.

Based on previous reports in the literature, additional antigenic epitopes were ultimately obtained, as shown in Table 5.

The predicted and reported epitopes were connected in a specific sequence, in which KK Linker was used to link the B-cell epitopes and GPGPG Linker was used to link the T-cell epitopes along with a rigid GPLS peptide to link B- and T-cell epitopes, as illustrated in the schematic diagram (Figure 5A). Through analysis with DNAStar Lasergene 11 (version 11) and DNAMan 6 (version 6) bioinformatics software, we screened and identified the optimal spatial arrangement and combination of highly immunogenic sites on the protein surface. The selected antigenic epitope-containing genes were subsequently concatenated using different linkers, followed by in-depth analysis and codon optimization. DNAStar software was used to analyze and evaluate the hydrophilicity, antigenicity index, surface distribution probability, and flexible region of each selected epitope. As shown in Figure 5B, epitopes in the ORF region demonstrate pronounced hydrophilic properties, robust antigenic potential, steric availability, and discrete conformational epitopes with structural integrity. The complete PRRS mRNA sequence designed in this study included the promoter sequence, 5′ UTR sequence, Kozak sequence, signal peptide coding sequence, ORF region, 3′ UTR sequence, and poly(A) sequence from the 5′ end to the 3′ end. The complete sequence was sent to Jiangsu Synthgene Biotechnology Co., Ltd. for the synthesis of mRNA-LNPs. Two micrograms of mRNA were transfected into 293T cells for 48 h, after which the expression was verified by IFA with PRRSV N pAb. As shown in Figure 5C, the synthesized mRNA was successfully expressed.

### 3.5. Evaluation of the Immunoprotective Effect of the PRRS mRNA Vaccine in Piglets

We evaluated the immunoprotective effect of the PRRS mRNA vaccine candidate. The vaccination and challenge protocol is illustrated in Figure 6A. As shown in Figure 6B, the PRRSV antibody level in the mRNA-vaccinated group was greater than that detected in the inactivated vaccine control group at 28 dpv. The results of the virus neutralization assays also indicate that the number of neutralizing antibodies detected in the PRRS mRNA vaccine group was greater than that detected in the inactivated vaccine group (Figure 6C).

To investigate whether the PRRS mRNA vaccine elicits protective immunity against heterologous viral exposure to JX021 PRRSV, piglets were challenged with 2 × 10^5^ TCID_50_ of HP-PRRSV JX021 at 28 dpv, and negative controls received DMEM. Rectal temperature, body weight, and clinical symptoms were monitored in surviving piglets in different groups at 14 dpc. As shown in Figure 7A, both the challenged and vaccinated groups presented fevers (≥40 °C) from 5 to 14 dpc. Among the three groups, the PBS group consistently presented the highest body temperature. In contrast, the increase in temperature caused by JX021 significantly decreased in the PRRS mRNA-vaccinated group. The PBS group experienced progressive weight loss, whereas both vaccinated groups maintained steady weight gain (Figure 7B). Clinical symptoms, including anorexia, depression, dyspnea, coughing, trembling, and paralysis, were observed in the PBS group. In contrast, the PRRS mRNA-vaccinated group presented significantly attenuated clinical manifestations (Figure 7C). The survival situation of piglets immunized with the mRNA vaccine was better than that of the inactivated vaccine immunization group and the PBS group (Figure 7D), indicating that the protective efficacy of the PRRS mRNA vaccine was better than that of the inactivated vaccine. Blood and serum samples were collected from all groups at 7, 10, and 14 dpc. Our results analyzed all surviving piglets in different groups. As shown in Figure 7E, seroconversion (S/P ratio > 0.4) occurred in the PBS group after 7 dpc, whereas the vaccinated groups maintained high levels of PRRSV-specific antibodies. Viremia peaked at 7 dpc, with the PRRS mRNA-vaccinated group showing the most significant reduction in viremia (*p* < 0.05) (Figure 7F). PRRSV RNA was detectable in nasal swabs collected at 3, 7, 10, and 14 dpc. Notably, the PRRS mRNA-vaccinated group presented the lowest viral load (*p* < 0.05) (Figure 7G). The PRRS mRNA-vaccinated group presented significantly reduced viral loads in tissues post challenge (*p* < 0.05) (Figure 7H). Importantly, the nontreated control piglets manifested no clinical symptoms throughout the study.

The PBS group presented severe pulmonary congestion, whereas the PRRS mRNA vaccine group presented significantly alleviated gross lesions. Histopathological examination was employed to assess the ability of the PRRS mRNA vaccine to protect against JX021-induced tissue damage. As shown in Figure 8, interstitial pneumonia was characterized by thickened alveolar septa and hemorrhage, as pointed out by the black arrow. Lymphocytic infiltration was observed in the PBS group, as pointed out by the red arrow. In contrast, lung sections from the groups receiving the PRRS mRNA and inactivated vaccines demonstrated markedly reduced pathological alterations. All the control groups maintained normal pulmonary morphology throughout the study.

In addition to evaluating clinical symptoms, viral detection, and pathological changes in the lungs, genes encoding proinflammatory cytokines, such as TNF-α, IL-1β, and IL-6, in tissues (lungs and submandibular lymph nodes) and serum were investigated to further evaluate the immunoprotective effect of the PRRS mRNA vaccine. According to our results, piglets presented remarkable increases in the levels of proinflammatory cytokines (TNF-α, IL-1β, and IL-6) (Figure 9). In contrast, the PRRS mRNA-vaccinated group presented significantly reduced levels of three cytokines (*p* < 0.05), outperforming the inactivated vaccinated group. These results indicate that the PRRS mRNA vaccine effectively suppresses PRRSV-induced cytokine storms.

## 4. Discussion

Porcine reproductive and respiratory syndrome virus (PRRSV) is a highly mutable RNA virus responsible for the widespread prevalence of PRRS in China [4]. Sublineage 1.8 (NADC30-like) PRRSV currently dominates epidemic strains [35], whereas sublineage 8.7 (HP-PRRSV) continues to pose substantial threats and economic losses to pig farms. Commercially available HP-PRRSV vaccines are prone to recombination with field strains, resulting in limited homologous protection and weak heterologous cross-protection, thereby complicating PRRSV prevention and control [36,37]. Consequently, developing novel vaccines (e.g., mRNA vaccines) against HP-PRRSV is imperative. An HP-PRRSV strain (JX021) was isolated from a pig farm in Jiangxi Province. Characterization confirmed its single-infection status and genetic stability. Sequencing of the ORF5 and NSP2 genes and whole-genome and recombination analyses revealed that JX021 belongs to sublineage 8.7 (HP-PRRSV) without recombination. We subsequently conducted pathogenicity analysis of JX021 PRRSV and reported that it can induce distinct clinical symptoms in piglets. Our results align with those of previous reports indicating that HP-PRRSV is highly pathogenic in piglets [38,39,40].

Owing to the suboptimal efficacy of traditional vaccines, PRRSV vaccine research has shifted toward novel platforms such as viral vector vaccines and nucleic acid vaccines [41,42,43]. Additionally, PRRSV structural proteins contain abundant antigenic epitopes that can serve as exogenous vaccine targets to increase immunogenicity. Among the epitopes, MHC I-restricted epitopes mainly present antigens to mediate cellular immunity and cannot directly induce antibody responses alone [42]. The major structural proteins of PRRSV, GP2a, GP3, GP4, GP5, M, and N, serve as critical protective antigens. Epitopes located on these proteins are recognized by host receptors to induce antibody responses [44]. Based on these theoretical foundations, we systematically screened B- and T-cell epitopes of PRRSV structural proteins (GP2a, GP3, GP4, GP5, M, and N) by combining immunological data with evidence from the literature. There are fundamental differences between human leukocyte antigens (HLAs) and swine leukocyte antigens (SLAs). However, in our immunoinformatics analysis, we comprehensively addressed species specificity in our design. Our target animal model employed swine. Moreover, the mRNA vaccine targeted the porcine virus, and all sequences used were derived from swine. Consequently, certain T-cell epitopes were predicted based on swine leukocyte antigen (SLA) models rather than human leukocyte antigen (HLA) systems. Certainly, future work will incorporate SLA tetramer assays and IFN-γ ELISpot to empirically validate epitope immunogenicity in swine. We subsequently designed a PRRS mRNA vaccine by linking selected epitopes via flexible linkers and synthesized the construct. In contrast to the PRRS mRNA vaccine developed by Zhou, which utilizes PRRSV GP5 and GP2-GP5-M as dual-target sequences for mRNA vaccine construction, our approach specifically targets critical epitopes of major structural proteins, thereby achieving more comprehensive antigenic coverage [43]. Following successful expression validation of the PRRS mRNA vaccine, we conducted immunization–challenge experiments. Piglets were vaccinated twice and then challenged with our isolated strain JX021. The results demonstrated that the JX021 challenge control group presented marked clinical symptoms (e.g., fever), viremia, and pulmonary lesions. In contrast, the PRRS mRNA-vaccinated group presented better survival situation, significantly attenuated clinical symptoms, and substantially reduced viral loads and pulmonary pathology compared with those of the challenge control group, which is consistent with the results of Ma and Chai [45,46]. These results indicate that the protective efficacy of the PRRS mRNA vaccine was greater than that of the inactivated vaccine. Two mRNA vaccines were evaluated in recent study [47]: one encoding full-length GP5-M fusion proteins (GP5-M) and another expressing the N protein fused to epitope segments of M/E (NMEpep). The GP5-M vaccine induced cellular immunity and conferred 60% survival post challenge. Similarly, our multiepitope mRNA-LNP vaccine—designed using immunoinformatics-guided epitopes—achieved an 80% survival rate. Both the GP5-M vaccine and our vaccine targeted PRRSV structural proteins and could reduce the viremia and lung pathology of the piglets, addressing key limitations of conventional and emerging platforms. However, our PRRS mRNA vaccine in this study also has limitations that warrant near-future refinements. First, the current epitope design targeting PRRSV structural proteins requires optimization through systematic and targeted screening. Furthermore, immunologically important nonstructural proteins (including NSP2 and NSP9) should be incorporated into future iterations [48,49]. Moreover, the absence of direct T-cell response metrics precludes definitive conclusions about cellular immunity mechanisms. Moreover, while reduced nasal and systemic viral loads suggest diminished transmission potential, empirical confirmation through contact challenge models is warranted. Future studies will integrate these analyses to comprehensively assess the vaccine’s field applicability. Finally, although the vaccine primarily targets HP-PRRSV, its cross-protective efficacy against emerging strains (e.g., NADC30-like variants) remains unverified. Future refinements will focus on evaluating the protective efficacy against broader subtypes of PRRSV strains.

## Figures and Tables

**Figure 1 microorganisms-13-01332-f001:**
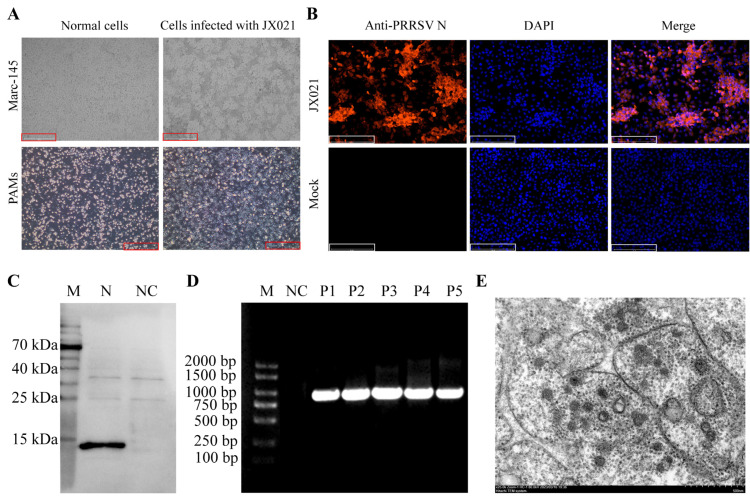
Identification of JX021 PRRSV. (**A**) CPEs were observed after PRRSV infection for 48 h in Marc-145 and PAM cells (scale bar = 100 μM). (**B**) Immunofluorescence assay (IFA) of Marc-145 cells infected with JX021 at 48 h. PRRSV was detected with PRRSV N pAb, and mock-treated Marc-145 cultures served as uninfected controls (scale bar = 500 μm). (**C**) PRRSV N protein synthesis was detected in Marc-145 cell monolayers, and the level of JX021 PRRSV was determined by Western blotting. (**D**) The stability of JX021 PRRSV after the first five passages was detected by PCR. DL2000 Plus DNA marker (Vazyme Biotech Co., Ltd., Nanjing, China) was used as a marker (Lane 1). The ddH_2_O was used as a negative control (Lane 2). P1 (Passage 1), P2 (Passage 2), P3 (Passage 3), P4 (Passage 4), and P5 (Passage 5) were the numbers of the first five passages of the JX021 strain (Lane 3–7). (**E**) The JX021 PRRSV virus was identified by transmission electron microscopy (scale bar = 500 nm).

**Figure 2 microorganisms-13-01332-f002:**
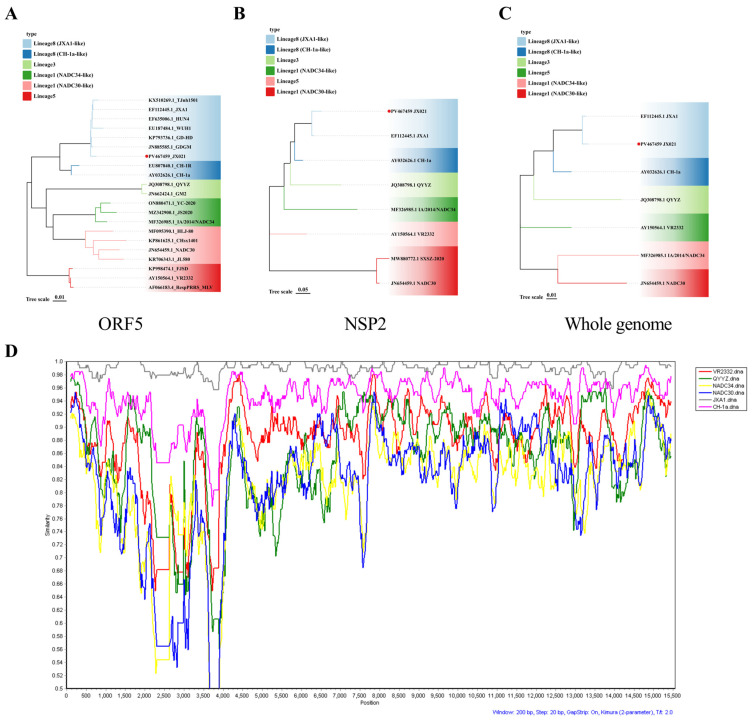
Evolutionary divergence and recombination profiling of JX021 were performed. (**A**–**C**) Phylogenetic analysis of JX021 PRRSV based on ORF5 (**A**), NSP2 (**B**), and the whole genome. (**C**) Phylogenetic reconstruction employing the neighbor-joining algorithm in MEGA11 was implemented, with JX021 demarcated by a red dot. (**D**) Recombination analysis of JX021 PRRSV using SimPlot (Version 3). JX021 was chosen as a query sequence, and the y-axis indicates the percentage identity in a sliding 200-bp window with 20-bp steps.

**Figure 3 microorganisms-13-01332-f003:**
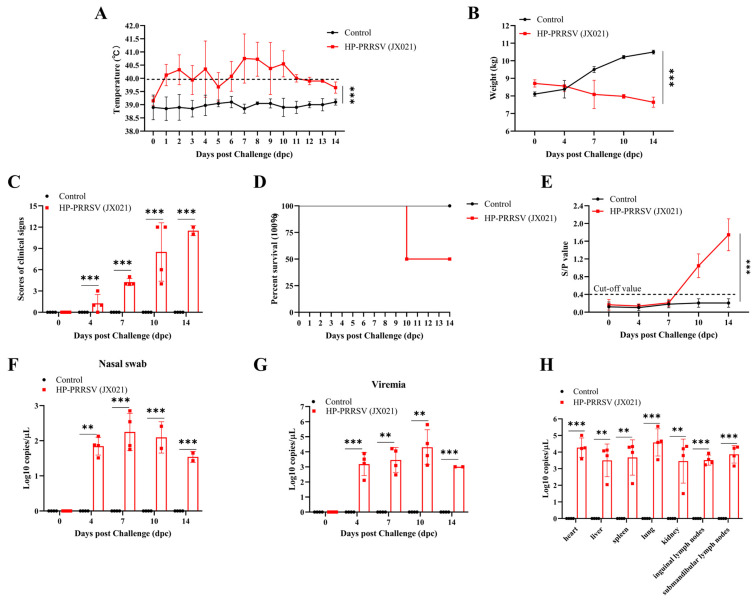
JX021 PRRSV exhibited marked virulence toward neonatal swine. (**A**,**B**) Dynamic alterations in core thermoregulatory parameters (**A**) and weight (**B**) were monitored in neonatal swine cohorts throughout a 14-day post-inoculation observation period, with a fever threshold set at 40.0 °C. (**C**) Variations in the clinical symptom scores of the piglets after challenge. (**D**) Te survival curves of piglets in the two groups. (**E**) Longitudinal monitoring of anti-nucleocapsid (N) immunoglobulin levels in peripheral blood was conducted at 0, 4, 7, 10, and 14 days post challenge (dpc) by using the commercial ELISA kit. A sample-to-positive (S/P) ratio of the piglets equal to or higher than 0.4 is considered positive (cut-off value), which was defined as seroconversion positivity. (**F**,**G**) Nasal swabs (**F**) and hematological monitoring (**G**) were conducted at infection onset and longitudinal intervals (0, 4, 7, 10, and 14 dpc) to quantify viremia kinetics. (**H**) Terminal necropsy at 14 dpc enabled assessment of the tissue-specific viral RNA burden via RT-qPCR in the JX021-infected cohort. When comparing between the two groups, Student’s *t*-test was used. Meanwhile, a one-way analysis of variance (ANOVA) was used when conducting comparisons of more than two groups. The datasets reflect the arithmetic mean ± standard deviation (vertical dispersion indicators) of each experimental group. ** indicates *p* < 0.01. *** signifies *p* < 0.001.

**Figure 4 microorganisms-13-01332-f004:**
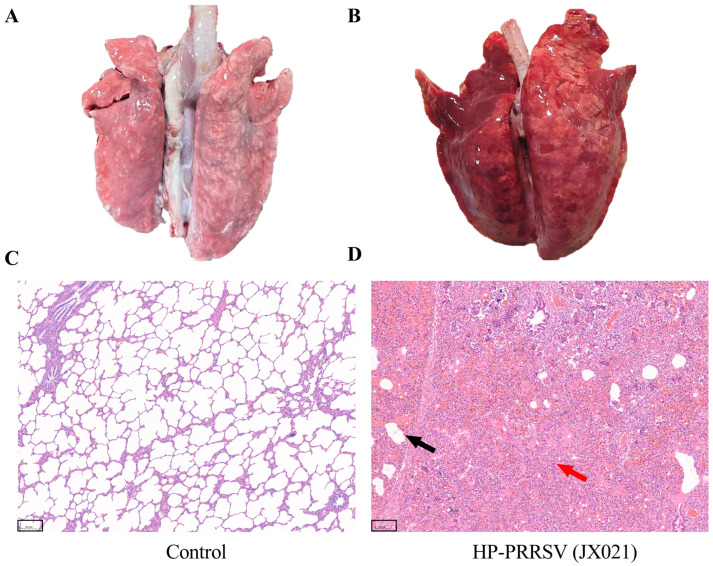
Representative lungs and pathological sections from each group (10×). (**A**,**C**) The control group (DMEM group); (**B**,**D**) the JX021 PRRSV-infected group. The lung tissue samples were immersed in 10% neutral buffered formalin (NBF), followed by paraffin embedding and microtome sectioning (4 μm) for histomorphological evaluation. Microtome-derived tissue sections were H&E-stained to facilitate the observation of micropathological changes (scale bar = 100 μm). The black arrow indicated the thickening of the alveolar septa. The red arrow indicated the infiltration of mononuclear cells.

**Figure 5 microorganisms-13-01332-f005:**
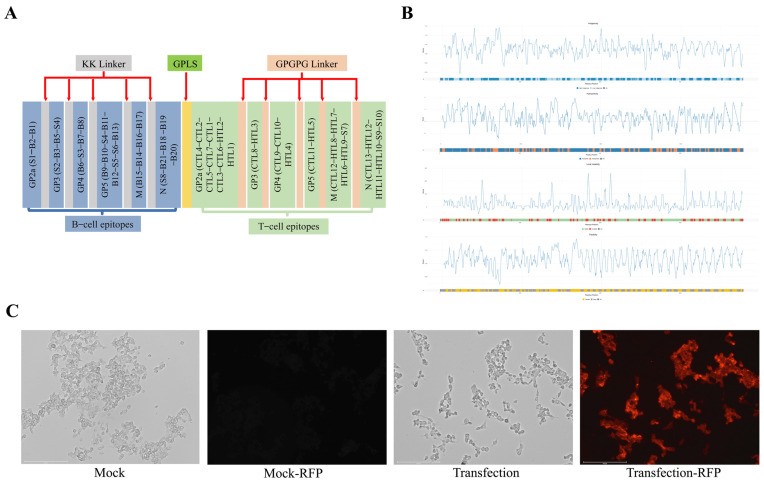
The design and expression of PRRS mRNA. (**A**) A schematic diagram of candidate ORF construction. (**B**) Evaluation of the hydrophilicity, antigenicity, surface accessibility, and independence of epitopes in the ORF region by R Studio (R Version 4.5). (**C**) The expression of PRRS mRNA. Immunofluorescence assay (IFA) of 293T cells transfected with mRNA. PRRSV was detected with a PRRSV N pAb, and the parental 293T cell line served as a nontransfected control (scale bar = 150 μm).

**Figure 6 microorganisms-13-01332-f006:**
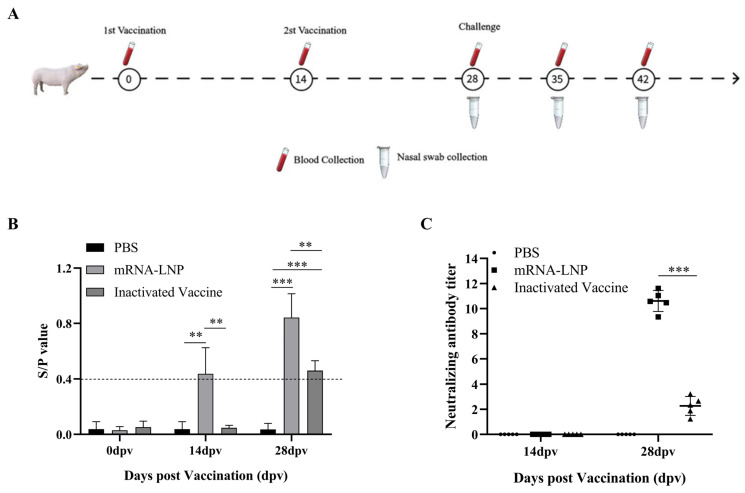
Immunogenic profiling of the PRRS mRNA vaccine. (**A**) Experimental design schematic delineating the vaccination regimen and synchronized viral challenge timeline in piglets. (**B**,**C**) Serum samples obtained at 14 and 28 dpv were analyzed to determine PRRSV antibodies by using the ELISA (**B**) and titers of neutralizing antibodies (**C**). PRRSV N protein levels were monitored at 0, 4, 7, 10, and 14 days post challenge (dpc) by using the commercial ELISA kit. A sample-to-positive (S/P) ratio equal to or higher than 0.4 is considered positive (cut-off value), which was defined as seroconversion positivity. Neutralizing antibodies were measured using an endpoint dilution reduction assay. For the analysis of neutralizing antibodies, non-parametric alternative was applied to compare the NAb response. Among two groups, the Mann–Whitney test was used. For more than two groups, the Kruskal–Wallis test was used. The median and 95% confidence intervals (CIs) were used. ** indicates *p* < 0.01. *** signifies *p* < 0.001.

**Figure 7 microorganisms-13-01332-f007:**
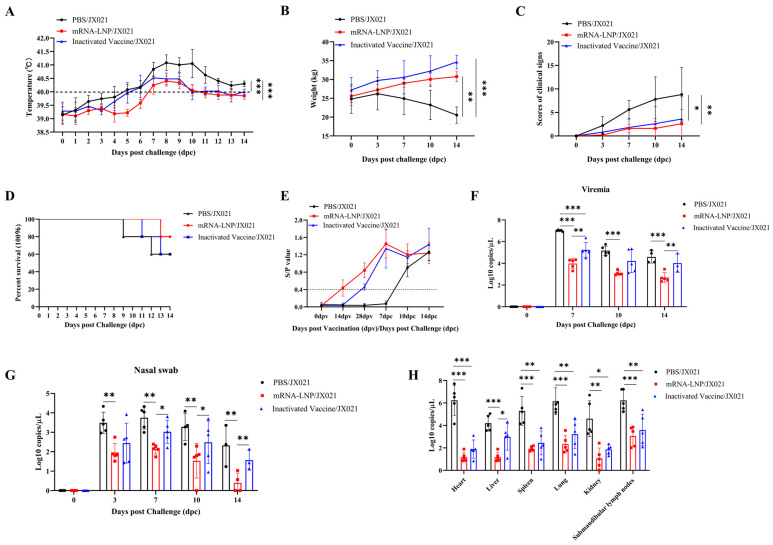
Evaluation of the protective efficacy of the PRRS mRNA vaccine against the JX021 challenge. (**A**,**B**) Porcine body temperature (**A**) and body mass (**B**) measurements after vaccination and challenge, with a fever threshold set at 40.0 °C. A one-way analysis of variance (ANOVA) was used when conducting comparisons of more than two groups. (**C**) Variations in porcine clinical manifestation scores after vaccination and challenge. A one-way analysis of variance (ANOVA) was used to analyze more than two groups. (**D**) The survival curves of the piglets in the three groups. (**E**) Variations in specific PRRSV N protein antibodies in the serum. Serum samples were collected at 0, 7, 10, and 14 dpc. A diagnostic threshold of seroconversion was established at an S/P of 0.4. (**F**,**G**) Nasal swabs (**F**) and blood samples (**G**) were collected at 0, 3, 7, 10, and 14 dpc to assess the quantified viral burden. Student’s t-test was used to conduct statistical analysis. (**H**) Tissue-specific viral quantification. When comparing between the two groups, Student’s *t*-test was used. Meanwhile, a one-way analysis of variance (ANOVA) was used when conducting comparisons of more than two groups. The values are expressed as group means ± SDs (error bars). * denotes *p* < 0.05. ** indicates *p* < 0.01. *** signifies *p* < 0.001.

**Figure 8 microorganisms-13-01332-f008:**
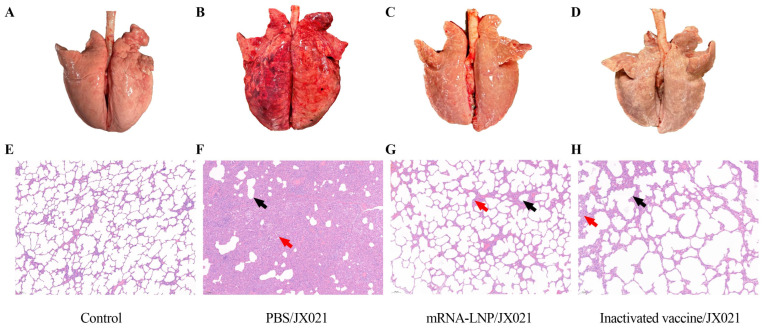
Pathology images of lungs and sections from each group (10×). (**A**,**E**) Lung (**A**) and pathological (**E**) sections from the control group (scale bar = 100 μm). (**B**,**F**) Lung (**B**) and pathological (**F**) sections from the JX021-challenged group (scale bar = 100 μm). (**C**,**G**) Lung (**C**) and pathological (**G**) sections from the PRRS mRNA-vaccinated group (scale bar = 100 μm). (**D**,**H**) Lung (**D**) and pathological (**H**) sections from the inactivated vaccinated group (scale bar = 100 μm). Pulmonary specimens were fixed in 10% neutral buffered formalin with subsequent paraffin embedding and microtome sectioning for histopathological evaluation. Colored arrows indicate different pathological changes. Histological sections were subsequently subjected to hematoxylin-eosin (H&E) staining to facilitate the observation of micropathological changes (scale bar = 100 μm). The black arrow indicated the thickening of the alveolar septa. The red arrow indicated the infiltration of mononuclear cells.

**Figure 9 microorganisms-13-01332-f009:**
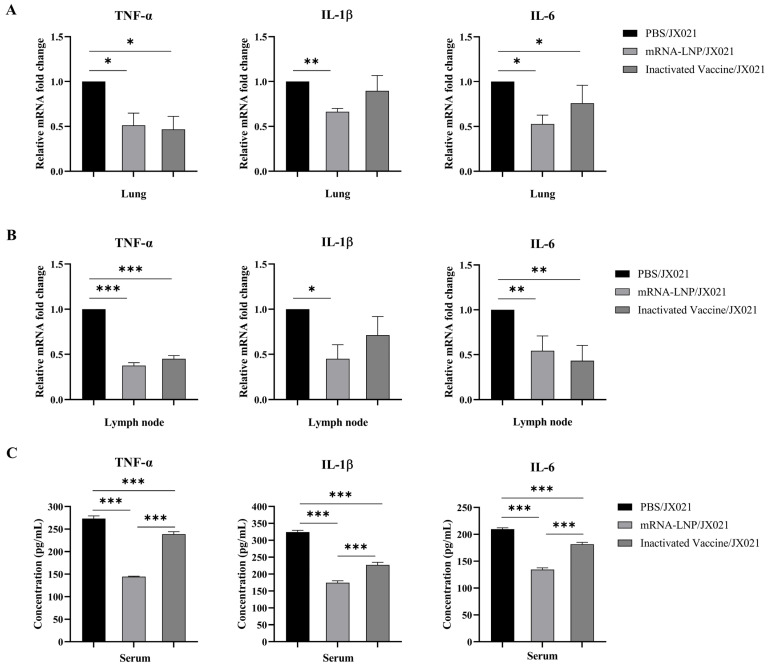
The PRRS mRNA vaccine significantly reduced TNF-α, IL-1β, and IL-6 expression in the lungs, lymph nodes, and serum. The lungs, lymph nodes, and serum were collected at 14 dpc. (**A**,**B**) The mRNA expression levels of TNF-α, IL-1β, and IL-6 in the lungs (**A**) and submandibular lymph nodes (**B**) were analyzed. Tissue-derived total RNA was isolated using TRIzol, and cDNA was synthesized by reverse transcription, after which the RNA was subjected to RT-qPCR. GAPDH served as the endogenous control, with the JX021-challenged group established as the reference standard. (**C**) Concentrations of TNF-α, IL-1β, and IL-6 in the serum were determined with commercial ELISA kits (Bioswamp Life Science Lab, Wuhan, China). When comparing between the two groups, Student’s t-test was used. Meanwhile, a one-way analysis of variance (ANOVA) was used when conducting comparisons of more than two groups. The group data are expressed as the means ± standard deviations, with error bars denoting variability. * denotes *p* < 0.05. ** indicates *p* < 0.01. *** signifies *p* < 0.001.

**Table 1 microorganisms-13-01332-t001:** Reference sequences of PRRSV.

Virus Strain	Accession No.	Origin
GM2	JN662424.1	China, 2011
JL580	KR706343.1	China, 2015
R98	DQ355796.1	China, 2006
SD16	JX087437.1	China, 2012
WUH1	EU187484.1	China, 2016
HuN4	EF635006.1	China, 2016
JS2020	MZ342900.1	China, 2021
JXA1	EF112445.1	China, 2016
CH-1a	AY032626.1	China, 1996
CH-1R	EU807840.1	China, 2008
HUB1	EF075945.1	China, 2016
QYYZ	JQ308798.1	China, 2011
CHsx1401	KP861625.1	China, 2014
NADC30	JN654459.1	USA, 2008
VR-2332	AY150564.1	USA, 1992
Lelystad virus	M96262.2	Netherlands, 1991
RespPRRS MLV	AF066183.4	USA, 1998
IA/2014/NADC34	MF326985.1	USA, 2017
IA/2015/NADC35	MF326986.1	USA, 2016
IA/2015/NADC36	MF326987.1	USA, 2016

**Table 2 microorganisms-13-01332-t002:** Vaccination and challenge experimental groups.

Groups	Number of Piglets	Vaccination	Vaccine Type	Injection Method and Dosage	HP-PRRSV Challenge	
					Strain	Method and Dosage
mRNA-LNP group	5	Yes	mRNA-LNP	150 μg, intramuscularly injected	JX021	2 × 10^5^ TCID_50_, intranasally inoculated and intramuscularly injected
Inactivated vaccine group	5	Yes	Inactivated vaccine	2 mL, intramuscularly injected	JX021	2 × 10^5^ TCID_50_, intranasally inoculated and intramuscularly injected
PBS group	5	No	-	2 mL PBS, intramuscularly injected	JX021	2 × 10^5^ TCID_50_, intranasally inoculated and intramuscularly injected
Sentinel group	5	No	-	2 mL PBS, intramuscularly injected	-	2 mL DMEM, intranasally inoculated and intramuscularly injected

**Table 3 microorganisms-13-01332-t003:** Predicted B-cell epitopes.

Protein	Name	Sequence	Start Position	End Position
GP2a	B1	GQAAWKQVVSEATLSR	123	138
	B2	SYASDWFAPR	49	58
GP3	B3	WCRIGHDRCSENDHDE	74	89
	B4	RVFRTSKPTPPQHQTS	211	226
	B5	HPEIFGIGN	123	131
GP4	B6	CKPCFSSSLSDIKTNT	22	38
	B7	HGDSSSPTIRKISQCR	55	70
	B8	LLHFMTPETMRWATVL	152	168
GP5	B9	TACCCSRLLFLWCIVP	7	22
	B10	NASNNNSSH	30	38
	B11	TDWLAQKFDW	53	62
	B12	VSTAGYYHGR	96	105
	B13	VLDGSAATPLTRVSAEL	180	196
M	B14	GFHPIAANDNHAFVVR	121	136
	B15	LAPAHHVESAAGFHPIAAND	110	129
	B16	RPGSTTV	137	143
	B17	VPGLKSLVLGGRKAVKQGVVN	148	168
N	B18	QSRGKGPGKKNRKKNP	35	50
	B19	LSSIQTAFNQGAGTCA	76	91
	B20	LPTQHTVRLIRATASP	106	121
	B21	NGKQQKKKKGNGQPVN	5	20

**Table 4 microorganisms-13-01332-t004:** Predicted T-cell epitopes.

Protein	Name	Sequence	Allele	Start Position
GP2a	CTL1	ASDWFAPRY	SLA-1: 0201	51
	CTL2	SQSPVGWWSY	SLA-1: 0501	41
	CTL3	HPLGVLWHH	SLA-1: 0701	93
	CTL4	SRNFWCPLL	SLA-3: 0101	21
	CTL5	SPVGWWSY	SLA-1: 0701	43
	CTL6	SQSPVGWWSY	SLA-2: 0301	151
	CTL7	SYASDWFAPRY	SLA-1-YC	49
	HTL1	LDQVFAIFPTPGSRP	HLA-DRB1*04: 01	187
	HTL2	VVAHFQHLAAIEAET	HLA-DRB1*11: 01	144
GP3	CTL8	HQVDGGNWF	SLA-2: 0201	171
	HTL3	GNVSQVYVDIKHQFI	HLA-DRB1*03: 01	130
GP4	CTL9	RTAIGTPVY	SLA-2: 0101	70
	CTL10	ETMRWATVL	SLA-1-CHANGDA	160
	HTL4	TPVYITITANVTDEN	HLA-DRB1*10: 01	75
GP5	CTL11	KFDWAVETF	SLA-1: 1301	59
	HTL5	KGRLYRWRSPVIVEK	HLA-DRB1*15: 01	149
M	CTL12	YSAIETWKF	SLA-2: 0101	86
	HTL6	GRKYILAPAHHVESA	HLA-DRB1*01: 01	105
	HTL7	LGRKYILAPAHHVES	HLA-DRB1*01: 01	104
	HTL8	LLGRKYILAPAHHVE	HLA-DRB1*07: 01	103
	HTL9	RKYILAPAHHVESAA	HLA-DRB1*07: 01	106
N	CTL13	ATEDDVRHHF	SLA-1: 0401	58
	HTL10	GKIIAQQNQSRGKGP	HLA-DRB4*01: 01	27
	HTL11	LGKIIAQQNQSRGKG	HLA-DRB5*01: 01	26
	HTL12	MLGKIIAQQNQSRGK	HLA-DRB5*01: 01	25

**Table 5 microorganisms-13-01332-t005:** Antigenic epitopes derived from the analogous region of the JXA1 strain reported.

Protein	Name	Sequence	Start Position	End Position	Types
GP2a	S1	RKAPPAHMARARGFKW [28]	43	56	B-cell epitope
GP3	S2	LEPGKSFWCRIGHDRCSEN [29]	67	85	B-cell epitope
GP4	S3	GDSSSPTIRK [30]	51	65	B-cell epitope
GP5-1	S4	SHIQLIYNL [31]	37	45	B-cell epitope
GP5-2	S5	LDTKGRLYRWR [31]	146	156	B-cell epitope
GP5-3	S6	GGKVEVEGHLIDLKRVV [31]	164	180	B-cell epitope
M	S7	KFITSRCRLCLLGRK [32]	93	107	T-cell epitope
N-1	S8	MPNNNGKQQKKKKGN [33]	1	15	B-cell epitope
N-2	S9	IAQQNQSRGKGPGKKNRKKNPEKPHFPLA [34]	30	58	T-cell epitope
N-3	S10	VRHHFTPSE [34]	63	71	T-cell epitope

## Data Availability

The original contributions presented in this study are included in the article. Further inquiries can be directed to the corresponding author. The nucleotide sequence was deposited in GenBank under the accession number PV467459 (PRRSV Porcine CHN JIANGXI 2022/8.21/E, complete genome).

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
