# Peer review of "Protective Efficacy of an mRNA Vaccine Against HP-PRRSV Challenge in Piglets"

_microorganisms, 2025, doi:10.3390/microorganisms13061332_

Round 1
Reviewer 1 Report
Comments and Suggestions for Authors
The authors isolated a NADC30-like PRRSV strain, named JX021, and characterized its virulence in pigs. In addition, using the sequence information of this isolate, the authors predicted several B and T cell epitopes and constructed an mRNA vaccine. They compared the efficacy of this vaccine with that of a killed vaccine derived from the homologous strain in pigs using a vaccination-challenge regimen and claimed that the efficacy of the mRNA vaccine was greater than that of the inactivated vaccine. Overall, the experiment is reasonably designed and the results are well presented. However, the manuscript still requires substantial revisions before it can be accepted.
-
Line 54: Arteriviridae should be italicized.
-
Line 74: in vivo should be italicized.
-
Line 177: The authors should explain the inactivated vaccine in more detail, including which isolate was used, how it was inactivated and reconstituted, and the viral dose in the final formulation.
-
Line 236: Figure 1 should be reordered to match the sequence in which each panel is discussed in the text.
-
Line 243: There is no scale bar in Figure 1A.
-
Line 283: There is no description of seroconversion in the Methods section.
-
Line 284: How was viremia determined? Please add methodological details in the Methods section.
-
Line 292: I suggest the authors indicate what statistical analysis was used in the figure captions (also for the other figures).
-
Line 312: I suggest adding colored arrows to indicate different pathological changes in the figures, along with corresponding descriptions in the figure legends (also for Fig. 8).
-
Line 332: Some T cell epitopes were predicted based on the HLA model rather than SLA. Since SLAs are significantly different from HLAs, the authors should explain why these epitopes were included in the mRNA vaccine design. This limitation should also be discussed.
-
Line 335: In Table 5, does GP6 refer to M? Does GP7 refer to N?
-
Line 335: What types of epitopes are listed (e.g., B cell, CTL, helper T cell)? Please add this information to the table.
-
Line 335: Are the epitope sequences identical to those of the JX021 strain?
-
Line 352: The Methods section describes the use of in-house polyclonal antibodies against N, but here monoclonal antibodies are mentioned for IFA. Please clarify this discrepancy.
-
Line 354: Figure 5B needs a more detailed figure legend, and the image resolution should be improved.
-
Line 354: How are the peptides ordered in the vaccine, especially when multiple peptides are derived from the same structural protein? It would be helpful to provide a more detailed schematic or the actual sequence of the mRNA vaccine.
-
Line 376: Should it be “homologous” instead? I believe the mRNA sequence was also derived from JX021, if I understand correctly.
-
Line 438: Please specify which lymph node was used for cytokine quantification and update this information in the manuscript.
-
Lines 467–468: The sentence is unclear. Why is “namely” used after GP2a? Please revise and provide a supporting reference.
-
Lines 469–470: MHC I-restricted epitopes do not necessarily induce antibody responses. Please revise this statement accordingly.
-
Lines 485–486: In this study, the mRNA vaccine appeared to be more effective than the homologous inactivated vaccine. I suggest the authors add further discussion on this point. For instance, the dose of the inactivated vaccine might have affected the results, as the quantities of antigen in the mRNA and inactivated vaccines are not directly comparable. Additionally, how would the mRNA vaccine compare with commonly used modified live virus (MLV) vaccines?
Reviewer 2 Report
Comments and Suggestions for Authors
The authors report the development of novel mRNA-based vaccines against PRRSV: piglets immunized with the mRNA vaccine had an 80% survival rate, whereas those in the inactivated vaccine immunization group and the challenge control group had a 60% survival rate. Compared with the inactivated vaccine group, the mRNA vaccine group presented reductions in viremia and lung lesions.
Have you consider to characterize the proinflammatory cytokine secretion profiling and perform the experiment thought in vitro methods, according to 3Rs principles?
Comments on the Quality of English Language
I recommend to revise the english
Reviewer 3 Report
Comments and Suggestions for Authors
The manuscript by Liu et al., presents the results of very extensive research work that goes from the isolation and genome sequence of a novel local isolate of PRRSV, the characterization of its virulence in pigs, a comprehensive prediction of B and T cell epitopes, the design of a T and B multi-epitope polypeptide (MEP), the synthesis of an RNA encoding for this MEP, and the evaluation of its immunogenicity and protective capacity from a challenge with the homologous viral strain.
The work is well designed and technically well executed, the cellular immunity is important to provide wide spectrum protection against PRRSV, therefore, the MEP strategy seems to be justified.
There are, however, several issues that authors must address before publication:
- The final sentences of the Introduction (lines 72-81) summarize the main results of the article. However, this practice is not methodologically sound, since the Introduction epigraph should be limited to provide a background and defining the objectives of the study. The authors must consider removing this information from the Introduction and limit to the definition of the objectives.
- Page 3, lines 97–108 describe the isolation of the viral strain. It is at this point that the new strain should be designated as JX021.
- Epigraph 2.6 on page 4 refers to two independent animal experiments. For clarity, I suggest splitting these into two separate experiments, 2.6.1 and 2.6.2, and providing more detail about the objectives of each experiment. Experiment 2.6.1 should study the virulence of strain JX021 in pigs, and experiment 2.6.2 should compare the new vaccine candidate with an inactivated vaccine.
- In this experiment, the names of groups 3 and 4 are confusing. Group 3 should be labelled the PBS group or the negative control group, not JX021-challenged control group, as groups 1, 2 and 3 were all challenged with the JX021 strain. Group 4, which is neither vaccinated nor challenged, should be named the 'sentinel' group.
- The authors should provide detailed information about the inactivated PRRSV vaccine used as a reference. Is it a commercial vaccine or an 'in-house' immunogen? Are there previous records of its efficacy? Was an adjuvant used? What was the administered dose? Is it homologous to the challenge virus? If not, this would be a significant disadvantage. Please provide a reference or describe the production procedures.
- The authors must provide more information about the final formulation of the RNA immunogen into lipid nanoparticles by Jiangsu Synthgene Biotechnology, or provide a reference to a detailed description of this procedure and the required composition to reproduce the experiment.
- Page 7, lines 245–246: “The stability of the first five passages of JX021 PRRSV was detected by PCR”. This sentence is inaccurate; I suggest the following: “The stability of JX021 PRRSV after the first five passages was assessed by PCR”.
- In the caption for Figure 1: Explain what M, N and NC, and P1-P5 mean.
- In the caption of Figure 1E, the bar is 500 nm, but no bar can be observed.
- On page 8, lines 252–253, it says, “The negative controls (n = 4) were infected with DMEM”. This sentence is incorrect, as DMEM is not infectious. Consider the following instead: “The negative controls (n = 4) were inoculated with DMEM”.
- Page 8, lines 289–290. “These data collectively suggest that JX021-induced PRRSV is highly pathogenic in piglets.” It is the virus that is pathogenic, not the virus-induced PRRSV. Consider the following: “These data collectively suggest that JX021 is highly pathogenic in piglets”.
- Page 9, figure 3E. Explain what the S/P values are and which method was used to measure the antibody response. The description of this method in the 'Materials and Methods' section is lacking.
- Page 13, line 358. Again, a bar is mentioned, but none is observed in the photograph.
- Page 13, line 360: “PRRS mRNA vaccine”: in this initial phase of research, it is more accurate to refer to the PRRS mRNA vaccine candidate or PRRS mRNA immunogen.
- Page 14, Figure 6B: This describes the results of the antibodies against PRRS. However, no information was provided in the “Materials and Methods” section about the procedures used in this ELISA. What antigens are used to measure the antibody response? Please include this information. In the figure captions, please explain the meaning of the S/P value on the x-axis, and use better contrast to differentiate the bars.
- Page 14. Figure 6C represents NAb titers. The caption should explain which scale is used. The NAb titer never follows a normal distribution; therefore, it is statistical meaningless to use the arithmetic mean and the standard deviation to represent the central tendency and variability. Instead, the median and 95% confidence intervals should be used. However, since the graphic represents all individual values, using the median will be sufficient. Consequently, instead of ANOVA, a non-parametric alternative must be used to compare the NAb response among groups: the Kruskal Wallis test for more than two groups or the Mann-Whitney test for two groups. The figure captions must indicate which test did you used (This is valid for all figures in the manuscript expressing statistical significance.
- Page 14, line 380-382. “Among the three groups, the JX021-challenged group consistently presented the highest body temperature. In contrast, the increase in temperature caused by JX021 significantly decreased in the PRRS mRNA-vaccinated group” The sentence is confusing. I suggested changing the names of the groups, since all groups except group 4 were challenged with the JX021 virus.
- Page 15. Figure 7D. Indicate in the figure caption which statistical test did was used in each case.
- Page 14 lines 387/390. This is the most important result, since survival is the primary goal of a vaccine. The authors wrote that “The survival rate of the PRRS mRNA-vaccinated group was 80% post-challenge, whereas it was 60% in both the inactivated vaccine and challenged groups (Figure 7D), indicating that the protective efficacy of the PRRS mRNA vaccine was better than that of the inactivated vaccine” However, in this experiment the survival rate at 14 days was the same for the inactivated vaccine as for the control group (60%). This means that with n=5, 2 pigs died and 3 survived. The RNA immunogen group shows 80% survival (4 pigs survived and one died). These differences cannot be statistically significant; thus, despite a better immune response, weight gain, fewer clinical signs, and a lower viral load, the conclusion is that the RNA vaccine is as ineffective as the inactivated vaccine in protecting against death by PRRSV.
- Figure 8 shows the results of the histopathological examination of the lungs. Only one animal per group is shown. How were these pigs selected since each group is heterogeneous.
- As previously mentioned, the T cell response appears to be very important for protection against PRRSV. However, the authors did not conduct any experiment to evaluate T helper or CTL responses in the animals.
- Page 18, lines 449-450. “highly mutagenic RNA virus” is inaccurate since the term mutagenic refers to a chemical compound that induces mutagenesis, “highly mutable” would be a better term.
- The discussion chapter could be improved by comparing the results with those of previous studies and analyzing the limitations of the study, such as the lack of evaluation of the T cell response and the lack of evidence of protection against mortality. It would also be interesting to discuss whether the observed reduction in the viral loads could be sufficient to limit the viral transmission, etc.
Comments on the Quality of English Language
A revision of the english language could be helpful
Reviewer 4 Report
Comments and Suggestions for Authors
I accept this manuscript as the first study of the possibilities of using mRNA vaccine for the prevention of PRRS in pigs. From a technical point of view, the manuscript is very well done. The Introduction provides a very good justification for conducting this study, the methods are well described, the results are very well presented, the discussion is also well done, and the conclusions are based on the results obtained. The results show that the protective efficacy of the PRRS mRNA vaccine is greater than that of the inactivated vaccine. The survival rate of the group vaccinated with PRRS mRNA is 80% after infection, while it is 60% for the group treated with the inactivated vaccine. This is also confirmed by the immunological and pathohistological studies performed. In parallel, I have some remarks:
- The study was conducted with a very small number of piglets, and therefore, it is not possible to draw radical conclusions for the mass use of the vaccine.
- However, the PRRS mRNA vaccine in this study also has limitations that require refinement in the near future:
- The current epitope design targeting PRRSV structural proteins requires optimization through systematic and targeted screening.
- Immunologically important non-structural proteins (including NSP2 and NSP9) should be included in future integrations.
- Although the vaccine primarily targets HP-PRRSV, its cross-protective efficacy against emerging strains (e.g., NADC30-like variants) remains unconfirmed. Future refinements should focus on evaluating protective efficacy against broader subtypes of PRRSV strains.
- No side effects have been reported with the PRRS mRNA vaccine, and these are likely to be similar to mRNA vaccines against COVID-19 in humans.
Round 2
Reviewer 1 Report
Comments and Suggestions for Authors
Line 136 Table 1: Replace “America” with “USA” for clarity, as “America” is not specific to a single country.
Line 186, line 214 and Table 2: The authors state that pigs were infected via either intranasal or intramuscular injection. It is essential to clarify whether all animals were challenged using the same route, as intranasal and intramuscular injections can elicit distinct immune responses due to differences in antigen presentation. In Table 2, all pigs are described as being challenged intramuscularly, which contradicts the description in the Methods section (lines 186 and 214). Please resolve this inconsistency.
Line 220, Table 2: Was the JX021 inoculum adjusted to 2 mL for each pig? If so, please explicitly state this in the table or methods.
Line 234: Specify the duration of serum incubation at 56°C.
Line 235: Provide the ×g value that corresponds to the 5,000 rpm centrifugation step.
Line 230: The description of the virus neutralization assay is incomplete. Please explain how the neutralizing titer was determined. The current sentence ending with “replaced for 5–7 days” is vague and lacks a clear endpoint measurement.
Line 406, Table 4: The use of the HLA binding model for SLA epitope prediction is acknowledged in the Methods section. However, this methodological limitation should also be discussed explicitly in the Discussion section.
Line 409, Table 5: I recommend clarifying in the table caption that the sequences listed were derived from the analogous region of the JXA1 strain, as reported in the cited literature.
Discussion: I suggest the authors include a comparison with the recent study on PRRSV mRNA vaccination (Mou, C., Zhao, X., Zhuo, C., He, Q., Xu, M., Shi, K., ... & Chen, Z. (2025). The mRNA Vaccine Expressing Fused Structural Protein of PRRSV protects piglets against PRRSV challenge. Veterinary Microbiology, 110534). This comparison would enhance the discussion of the current study’s novelty and relevance.
Reviewer 3 Report
Comments and Suggestions for Authors
The authors have correctly addressed most of my previous comments, however, there are two aspects that still need an improvement:
- Page 2. Lines 74-78. The paragraph was added at the end of the Introduction following my previous suggestion. However, the redaction of this paragraph must be improved and the objectives of the study must be more clearly defined:
- to isolate and sequence a novel PRRSV strain from China and to characterize its virulence in pigs to be used in challenge experiments.
- to predict B and T cell epitopes in PRRSV proteins.
- to construct an RNA vaccine candidate against PRRSV encoding a multi-epitope polypeptide, including B an T cell epitopes and.
- to evaluate the immunogenicity of the RNA vaccine in comparison with a commercial inactivated vaccine against PRRSV and its capacity to protect against a viral challenge.
- Page 14, lines 485-488 and page 18, lines 608-609. The sentences still need to be rephrased. As I mentioned in my previous review if statistical differences cannot be demonstrated due to the small sample sized of the study, this assertion is incorrect. This limitation should be clearly established, as should the fact that, although the survival rate was slightly higher, (only one pig), no statistical significance can be yet demonstrated regarding protection against mortality.
